# Electronic Janus lattice and kagome-like bands in coloring-triangular MoTe$_2$ monolayers

Le Lei[1,2,7], Jiaqi Dai[1,2,7], Haoyu Dong[1,2,7], Yanyan Geng[1,2], Feiyue Cao[1,2], Cong Wang [1,2], Rui Xu[1,2], Fei Pang [1,2], Zheng-Xin Liu [1,2], Fangsen Li [3,4], Zhihai Cheng [1,2] ✉, Guang Wang [5,6] ✉ & Wei Ji [1,2] ✉

Polymorphic structures of transition metal dichalcogenides (TMDs) host exotic electronic states, like charge density wave and superconductivity. However, the number of these structures is limited by crystal symmetries, which poses a challenge to achieving tailored lattices and properties both theoretically and experimentally. Here, we report a coloring-triangle (CT) latticed MoTe$_2$ monolayer, termed CT-MoTe$_2$, constructed by controllably introducing uniform and ordered mirror-twin-boundaries into a pristine monolayer via molecular beam epitaxy. Low-temperature scanning tunneling microscopy and spectroscopy (STM/STS) together with theoretical calculations reveal that the monolayer has an electronic Janus lattice, i.e., an energy-dependent atomic-lattice and a Te pseudo-sublattice, and shares the identical geometry with the Mo$_5$Te$_8$ layer. Dirac-like and flat electronic bands inherently existing in the CT lattice are identified by two broad and two prominent peaks in STS spectra, respectively, and verified with density-functional-theory calculations. Two types of intrinsic domain boundaries were observed, one of which maintains the electronic-Janus-lattice feature, implying potential applications as an energy-tunable electron-tunneling barrier in future functional devices.

Two-dimensional (2D) materials received increasing attention due to their exotic electronic and optical properties[1-13]. Recently, it comes into another intriguing property of 2D materials that engineering of their rich polymorphs showing diverse properties for wide applications[14-16]. Polymorph refers to the concept that a given composition with a variety of different crystal structures, including single-element materials and compounds. For instance, borophene possesses a highly polymorphic characteristic. It exhibits many atomic structures due to the complexity of bonding motifs[17], in which a series of exotic properties, including massless Dirac fermions[18] and 1D nearly free-electron states[19], were found. Additional states, such as superconductivity[20], charge density wave (CDW)[21-23], and nontrivial topological states[24] were recently found in layered transition metal dichalcogenides (TMDs), offering a particular platform to investigate

[1]Beijing Key Laboratory of Optoelectronic Functional Materials & Micro-nano Devices, Department of Physics, Renmin University of China, Beijing 100872, China. [2]Laboratory of Quantum State Construction and Manipulation (Ministry of Education), Renmin University of China, Beijing 100872, China. [3]Vacuum Interconnected Nanotech Workstation, Suzhou Institute of Nano-Tech and Nano-Bionics, Chinese Academy of Sciences, Suzhou 215123, China. [4]School of Nano-Tech and Nano-Bionics, University of Science and Technology of China, Hefei 230026, China. [5]Department of Physics, College of Sciences, National University of Defense Technology, Changsha 410073, China. [6]State Key Laboratory of Low-Dimensional Quantum Physics, Department of Physics, Tsinghua University, Beijing 100084, China. [7]These authors contributed equally: Le Lei, Jiaqi Dai, Haoyu Dong. ✉e-mail: zhihaicheng@ruc.edu.cn; wangguang@nudt.edu.cn; wji@ruc.edu.cn

fundamental condensed matter physics in the two-dimensional limit. Monolayer TMDs were successfully fabricated in many polymorphic phases, such as the hexagonal (1H), octahedral (1 T) and monoclinic (distorted octahedral) (1 T′)[25] phases, showing phase-related properties. For example, the monolayer (ML) 1T-NbSe$_2$ exhibits a √13 × √13 CDW order and a correlated magnetic insulating state[26], but its 1H counterpart possesses a 3 × 3 CDW order and superconductivity[20].

Mirror twin boundaries (MTBs)[27,28] were demonstrate to be another strategy to introduce additional exotic electronic states in chalcogen-deficient 1H-MoS$_2$[29], -MoSe$_2$[27], and - MoTe$_2$[30] monolayers. Infinite-length MTBs were theoretically revealed to be metallic and show a high density of states (DOS)[31] near the Fermi level ($E_F$). However, they usually exhibit gaps near $E_F$ in STS spectra of those monolayers at low temperature, ascribed to the formation of charge orders, like Peierls-type CDW[32,33] or Tomonaga-Luttinger liquid[34,35]. The MTBs, in form of chalcogen-sharing lines, develop in three equivalent zigzag (ZZ) directions of the TMD monolayer lattice. This three-fold equivalence enables the MTBs to form triangular structures and dense networks[27], which could serve as block units for potentially building well-defined, like kagome[36,37], or coloring-triangle[38] lattices. Although it poses a huge challenge to experimental realization, a TMD layer consisting of ordered and uniformly sized MTB triangles, namely an MTB-triangle lattice[39,40], could be a TMD phase exhibiting a well-defined lattice symmetry. Therefore, this strategy allows the expansion of the family of polymorphic TMD phases, which are essential for exploring exotic electronic states in the 2D limit.

In this work, we constructed a coloring-triangle (CT) lattice in a MoTe$_2$ (CT-MoTe$_2$) monolayer comprised of uniform-sized and orderly arranged MTB triangular loops and normal MoTe$_2$ domains embedded among MTBs. This CT-MoTe$_2$ monolayer was theoretically proposed and experimentally prepared using a controllable annealing process to an as-grown MoTe$_2$ monolayer. The geometric and electronic structures of CT-MoTe$_2$ were measured using scanning tunneling microscopy/spectroscopy (STM/STS) and verified with first-principles calculations, which reveal an electronic Janus lattice showing two energy-dependent lattices. Further STS measurements in CT-MoTe$_2$ show two prominent peaks near the Fermi level ($E_F$), which are related to two electronic bands inherently existing in CT lattices, as observed in our theoretical calculations. Furthermore, we found a domain boundary in CT-MoTe$_2$, which becomes invisible in certain energy windows. In other words, it behaves like an energy-tunable barrier for electrons flowing through. This work sheds considerable light on the identification of more complicated but uniform polymorphs of TMD monolayers which exhibit exotic electronic phenomena in 2D systems.

## Results
### Monolayer MoTe$_2$ and MTBs
Figure 1a shows a typical STM topographic image of the MoTe$_2$ sample after post-annealing of an epitaxially grown monolayer on a HOPG substrate at ~513 K. The accompanying STM current image (Fig. 1b) clearly manifests the coexisting of 1H- and 1 T′-MoTe$_2$ phases[25]. A substantial portion of the 1H regions is covered by high-density mirror-twin boundaries (MTBs), as shown in Fig. 1b and Supplementary Fig. 1, forming various dense MTB networks and/or triangles in different sizes. Figure 1c, d show representative high-resolution topography images of the MTB structures, which appear as single (Fig. 1c) and double (Fig. 1d) bright stripes for the empty and occupied states, respectively. The d$I$/d$V$ spectra of 1H- and 1 T′-MoTe$_2$, as shown in Fig. 1e and Supplementary Fig. 2d, reveal a semiconducting bandgap of ~1.9 eV, consistent with a previous value of 2.01 eV[33], and a semimetallic gapless feature[41], respectively. A distinct narrow U-shaped gap around $E_F$ was observed in the d$I$/d$V$ spectrum acquired on the MTB structure (Fig. 1f), where two sharp peaks residing at the two sides of the gap, ascribed to a Peierls-type CDW[32,33] or Tomonaga-Luttinger liquid state[34].

The interconnected MTBs of the MoTe$_2$ sample form triangular loop structures surrounding the pristine 1H-MoTe$_2$ domains (Fig. 1d) in which the sizes of the triangles are randomly distributed. Figure 1g–i illustrate the atomic models of three MTB triangles in sizes of $N$ = 1, 3, and 6, where N represents the number of Te$_2$ units (red in Fig. 1g–i) in the 1H-MoTe$_2$ domain. Here, an MTB triangle (highlighted by orange dotted triangles in Fig. 1g–i) contains a Mo-terminated triangular domain (MTTD) of pristine 1H-MoTe$_2$ (highlighted by red hashed triangles in Fig. 1g–i, Te$_{MTTD}$-R atoms are highlighted by red balls) and its surrounding MTB loops (Te lines). The size of MTTD is governed by the chalcogen deficiency, as demonstrated in the literature[30]. It was thus an effective route to control the size of MTB triangles that controllably post-growth removal of chalcogen atoms, which was recently achieved by annealing the sample at a certain temperature during a certain period of time[27,42,43]. In the high MTB density limit, the smallest MTB triangle ($N$ = 1, Fig. 1g) dominantly presents, which, most likely, has a sufficiently large condition window for experimental realization and is expected to host exotic electronic states. We thus use $N$ = 1 MTB triangles (MTTDs) for illustration in our theoretical proposal.

### Design and construction of CT-MoTe$_2$ phase
Figure 2a depicts an ordered triangular lattice of the smallest MTTDs, highlighted using red dashed triangles, arranged in a corner-to-edge manner, while blue solid triangles displayed in Fig. 2b highlight a hexagonal lattice of the smallest MTTDs oppositely oriented to the red triangle and assembled in a corner-to-corner manner. Their interstitial regions are filled with the Te lines of MTBs (orange dotted lines in Fig. 2a and b). These MTTDs, together with the MTB loops, form a novel polymorphic MoTe$_2$ phase, the lattice model of which was recently proposed in theory as the coloring-triangular (CT) lattice (Fig. 2c), hosting kagome-like electronic bands[38]. Thus, we denote this phase as the CT-MoTe$_2$ phase. The CT lattice is a variant of the kagome lattice which is an interesting and well-defined lattice and consists of many sets of two Dirac bands and one flat band. The relationship between the kagome and CT lattices was illustrated in Supplementary Fig. 3.

Density functional theory calculations were further carried out to elucidate the existence of the CT-MoTe$_2$ monolayer. Figure 2d shows the fully relaxed atomic structure of the CT-MoTe$_2$ monolayer, which is, as we proposed, comprised of oppositely oriented $N$ = 1 MTTDs (red and blue triangles) being separated by the MTB loops (Te lines, highlighted by orange dotted triangle). Such an arrangement yields a triangular superlattice with a lattice constant of 12.66 Å (black dashed rhombuses in Fig. 2a–c. However, those central Te atoms of the MTTDs, regardless of their orientations, spatially reside in a smaller triangular lattice with a lattice constant of 7.31 Å. We denote it as the Te pseudo-lattice in the CT-MoTe$_2$ monolayer that its periodicity is expected to show under certain selected energies. This smallest MTB monolayer shares the identical geometry and chemical ratio with the recently found Mo$_5$Te$_8$ monolayer[39].

Figure 2e shows the electronic band structure of the CT-MoTe$_2$ monolayer, in which a 0.09 eV energy bandgap opens at the Fermi level. An on-site Coulomb energy $U$ = 1.5 eV is mandatory to obtain this bandgap, suggesting correlated electronic characteristics of the CT-MoTe$_2$ monolayer. Details of $U$-dependence on the electronic band structures were discussed in Supplementary Fig. 4. There are 12 bands residing near $E_F$, which are categorized into four sets of kagome bands, denoted CT1 to CT4 and coloured in green, blue, violet, and grey, respectively, according to symmetry analysis shown in Supplementary Table I. The irreducible representations (irreps, IRs) of those states at the Γ point were found to be (Γ1, Γ5), (Γ6, Γ2), (Γ3, Γ6), and (Γ5, Γ1), respectively. Thus, sets CT1 (CT2) and CT4 (CT3) are connected through a mirror symmetry operation σ$_h$ that Γ1 and Γ5 of bands in sets CT1 and CT4 have character 1, while Γ2, Γ3 and Γ6 have character -1 for

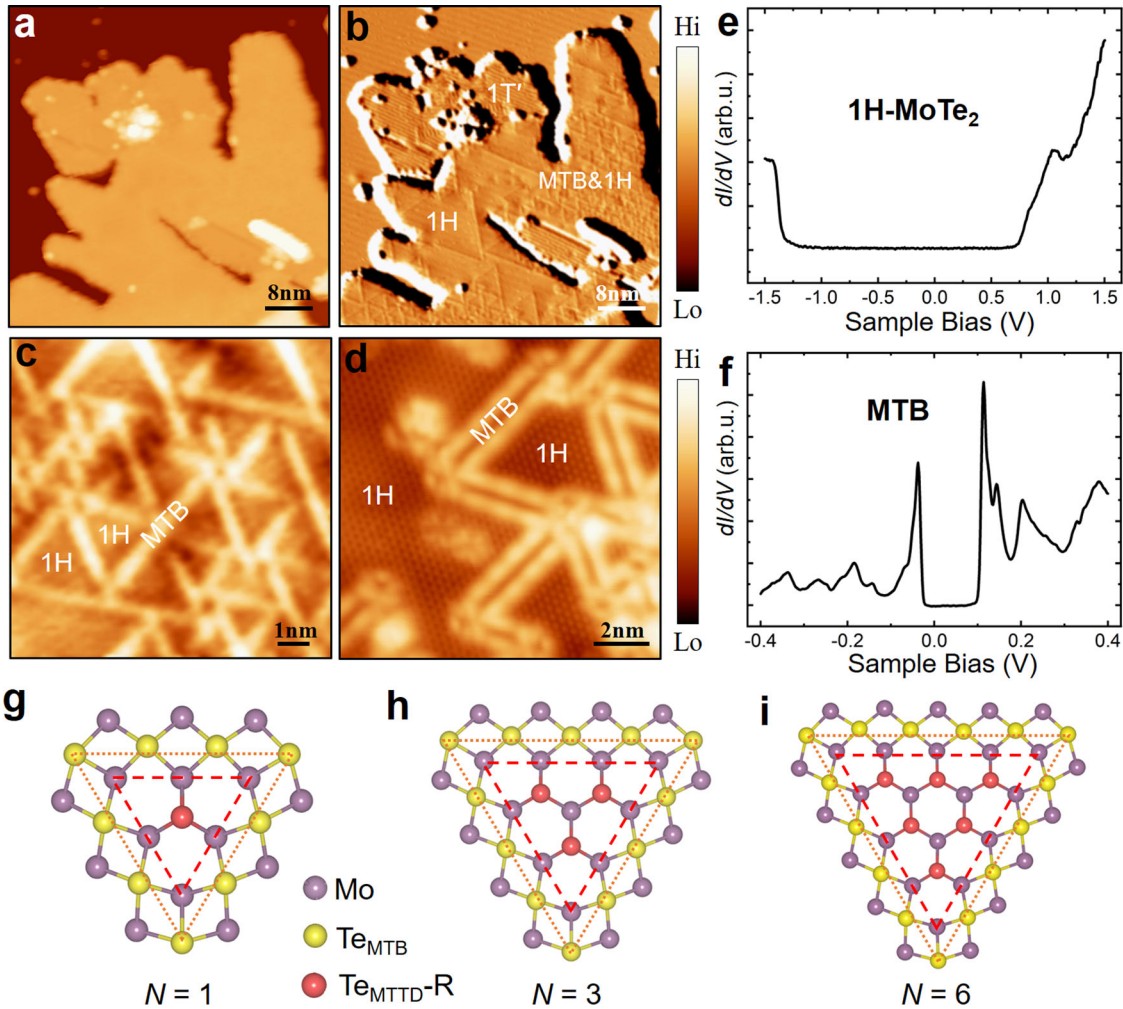

**Fig. 1 | Morphology and polymorphs of monolayer MoTe₂. a, b** Large-scale scanning tunneling microscopy (STM) topography (**a**) and current (**b**) images of the synthetic MoTe₂. Local 1H- and 1 T′-MoTe₂ phases, and mirror-twin boundaries (MTB) with 1H phase are labelled as 1H, 1 T′ and MTB&1H, respectively. **c** Magnified STM topographic image showing 1H-MoTe₂ domains and MTB networks. **d** Atomically resolved STM topographic image of MTB triangles. **e, f** Typical d*I*/d*V* spectra taken on the 1H-MoTe₂ domains (**e**) and MTB networks (**f**), respectively. **g**–**i** Illustration of the atomic models of MTB triangles of different sizes of N. The

red and yellow spheres represent Te atoms, the violet spheres represent Mo atoms. N represents the number of the Te₂ unit (red spheres) in the 1H-MoTe₂ domain. Red hashed triangles outline Mo-terminated triangular domains (MTTD), while their associated MTB loops are highlighted using orange dotted triangles (hereinafter). Hi and Lo represent High and Low in the color scalebar of STM images herein. Scanning parameters are (**a, b**) $V = 2.6$ V, $I = 100$pA, 52 nm × 52 nm; (**c**) $V = 2.0$ V, $I = 70$pA, 10 nm × 10 nm; (**d**) $V = -1.1$ V, $I = -80$pA, 12 nm × 12 nm.

bands in sets CT2 and CT3. A similar mirror symmetry connection was observable in bands CT1 to CT4 at the K point.

The flat band of CT2 resides at approximately 0.05 eV (over $E_F$, hereinafter), while the all three bands of set CT3 were observable from 0.19 to 0.28 eV in the density of states (DOS) plotted in Fig. 2f, which were denoted states t-S1 and t-S2, respectively. All bands in the green set (CT1) are more dispersive than those in sets CT2 and CT3. The two Dirac-like bands of CT1 show a gap from −0.31 to −0.43 eV, one of which, together with the dispersive "nominal flat band" are observable from −0.02 to −0.37 eV, denoting state t-S0 in Fig. 2g. State t-S3 represent part set CT4 where both the "nominal flat band and the Dirac bands are, again, highly dispersive". The bands of sets CT1 (green) and CT2 (blue) residing around $E_F$ more clearly show the kagome-like features in Fig. 2g and h, which primarily constitute of Te *p* states and Mo *d* states. Each set contains a nearly or nominal flat-band and two highly dispersive bands showing Dirac-like behaviors around the K point, although a bandgap opens at K for the CT1 set. Details of these kagome-like bands were discussed in Supplementary Figs. 5 and 6.

We use these two sets for spatial illustration of the CT lattice. We visualized their wavefunction norms of the −0.44 eV (CT1-A) state at

the K point and the degenerated states above (−0.32 eV, CT1-B) in Fig. 2i and j, respectively. They, as highlighted using triangles, both show the pattern of the CT lattice. The CT1-A state is partially distributed on the $p_z$ states of Te$_{MTTD}$-R and Te$_{MTB}$−1 and −4 atoms, while the CT1-B state is comprised of the bonding state of the $p_{xy}$ orbitals of Te$_{MTB}$−2 and −3 and the $p_{xy}$ state of Te$_{MTTD}$-B. By this means, CT1-A and CT1-B are energetically and spatially separated. The band structure (Fig. 2h) and visualized wavefunction norms (Fig. 2k and l) of set CT2 exhibit comparable patterns, namely a set of kagome-like bands and CT-symmetry appeared wavefunction norms. However, the contribution from Te$_{MTTD}$-R nearly eliminates in visualized wavefunction norm of the CT2-A state (0.05 eV) at the K point and those for the Te$_{MTB}$−1 and −4 atoms are from $p_{xy}$ orbitals (Fig. 2k). Unlike CT1-B, CT2-B is partially constituted of the *anti*-bonding state of the $p_{xy}$ orbitals of Te$_{MTB}$−2 and −3 (Fig. 2l), which explains why the CT2 set sits at higher energy and is thus unoccupied. Both pronounced electronic contributions from Te$_{MTTD}$-R and Te$_{MTTD}$-B in CT1-A (Fig. 2i), most likely, result in a smaller apparent lattice period of the surface, termed the Te pseudo-sublattice, which is expected to be selectively visualized at certain energy windows. Given the exhibition of the two real-space

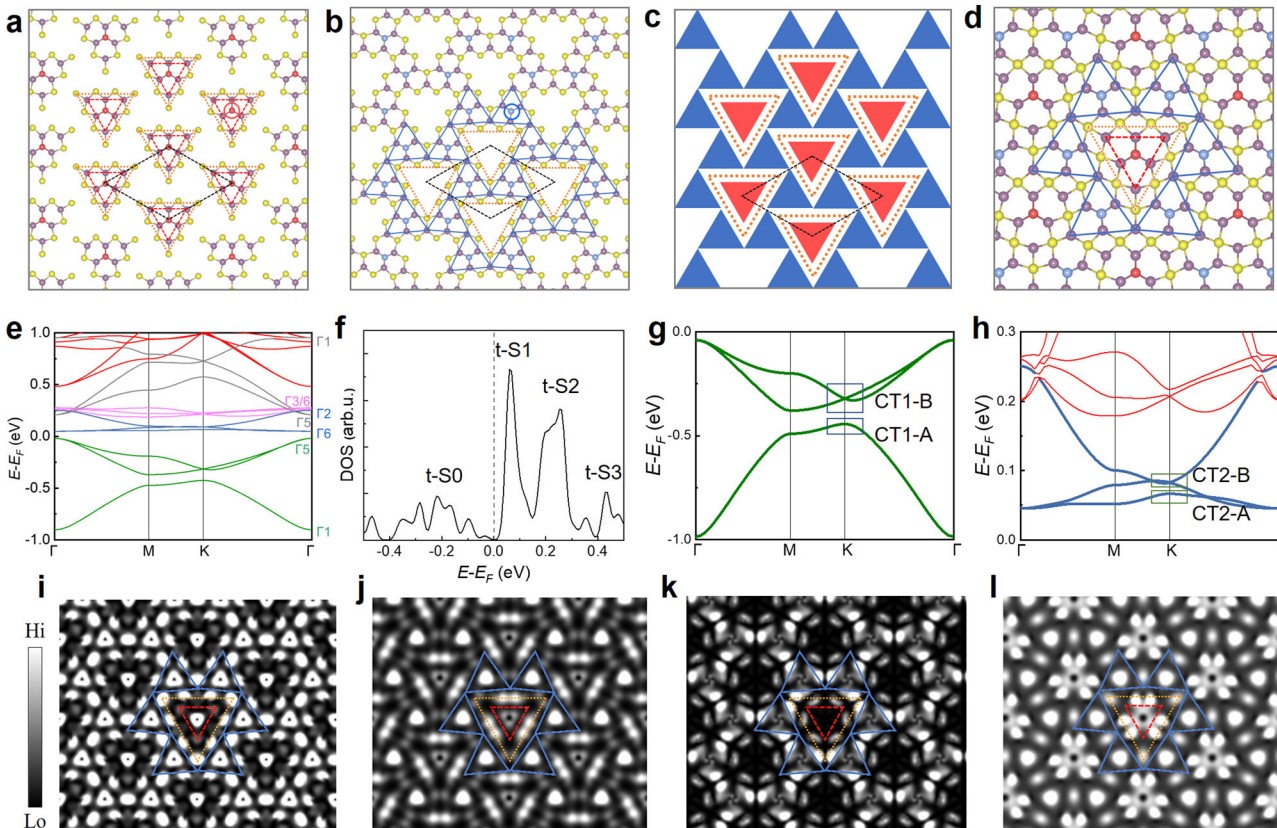

**Fig. 2 | Theoretical atomic and electronic structures of the coloring-triangle latticed MoTe₂ (CT-MoTe₂) phase. a** Illustration of the triangular lattice of the smallest MTTDs (highlighted by red dashed triangles) inside the associated MTB triangles (orange dotted triangles, hereinafter), which are arranged in a corner-to-edge manner. The black dashed rhombus indicates the supercell of the lattice (hereinafter). **b** Illustration of the hexagonal lattice of the smallest MTTDs (highlighted by blue triangles) among the associated MTBs, connected in a corner-to-corner manner. The central Te atoms in MTTDs are highlighted by the red (Te$_{MTTD}$-R) and blue (Te$_{MTTD}$-B) circles, respectively. **c** Schematic of the formed coloring-triangle (CT) lattice of CT-MoTe₂, composed of both triangular (**a**) and hexagonal

(**b**) lattice of the smallest MTTDs. **d** Illustration of the atomic structure of the CT-MoTe₂ phase. **e** Theoretical band structures of the CT-MoTe₂ monolayer, varying colors denote different sets of kagome-like bands and their irreducible representations (irreps) were also denoted. **f** Corresponding total density of states (DOS) of the CT-MoTe₂ monolayer. Four bands of t-S0, t-S1, t-S2 and t-S3 are shown in the DOS, where t represents theoretical. **g, h** Zoomed-in band structures of two sets of the CT bands display in green lines (CT1) and blue lines (CT2). **i–l** 2D contour of visualized wavefunction norms at the K point (see blue rectangles in **g** and **h**) for CT1-A (**i**), CT1-B (**j**), CT2-A (**k**), and CT2-B (**l**), respectively. The isosurface values were kept fixed at $2 \times 10^{-4}$ e Bohr$^{-3}$.

(quasi-)periodicities, the electronic characteristics of CT-MoTe₂ are "double-faced". We thus termed this feature as the electronic Janus lattice, in which term "Janus" focuses on electronic structures.

An interesting question is thus arisen that whether the kagome-like bands primarily originate from the lattice symmetry, as the in-plane anisotropy of atomic/molecular orbitals on sites of a non-CT or non-kagome lattice may also introduce kagome-like band structures[38]. Supplementary Fig. 6 shows the frontier orbitals of MTTD and MTB loops involved in forming bands in set CT1 to CT4. Most of these frontier orbitals are comprised of Mo $d$ and Te $p$ orbitals, including the out-of-plane $z$ component and the $C_3$ symmetrized in-plane component (Supplementary Fig. 5), which was demonstrated by a tight-binding (TB) $d$-orbital kagome lattice model intuitively[44]. Thus, the symmetry of these orbitals guarantees the origin from the lattice symmetry for those kagome-like bands observed in the present work.

### Synthesis and characterizations of CT-MoTe₂ phase

To experimentally prepare the CT-MoTe₂ monolayer, we post-annealed our samples shown in Fig. 1 at an even higher temperature of 616 K. Figure 3a shows a STM topographic image of the sample. Several wire-like features were observed at the edges of the monolayer islands, ascribed to Mo₆Te₆ nanowires (see Supplementary Fig. 7), clearly demonstrating a phase transition occurred at the edges under a

chalcogen-deficient condition[45]. Moreover, those domains inside the islands show an ordered phase, distinctly different from the 1H- or 1T'- phase, as locally resolved and marked with white dashed lines in the STM current images (Fig. 3b and Supplementary Fig. 8). A magnified current image (Fig. 3c) more clearly displays the features of the emergent phase, which are consistent with the proposed CT-MoTe₂ phase. A structural model of the CT-MoTe₂ monolayer was replotted in Fig. 3d where the lattice vectors of the two Janus electronic lattices are displayed in red (atomic-lattice) and blue (Te pseudo-sublattice).

Comparison of a series of bias-dependent experimental (Fig. 3i–h) and theoretical (Fig. 3i–l) STM images verifies that the prepared sample is the CT-MoTe₂ monolayer exhibiting the Janus electronic lattices. The experiment and theory are well consistent over a large range of bias voltages in terms of image appearance and apparent lattice periodicity. This series of images shows two apparent lattices, namely a larger one representing the atomic-lattice (red in Fig. 3e–h, a smaller one showing the electronic Te pseudo-sublattice (blue in Fig. 3e and h). The atomic-lattice was exclusively imaged at −0.4 (Fig. 3f) and +0.2 V (Fig. 3g), showing apparent tri-spots features in the 12.2 Å lattice (indicated by the red vectors in Fig. 3e–h). At higher bias voltages, namely −1.3 and +1.5 V, the Te pseudo-sublattice was imaged as bright spots (Fig. 3e) or black pits (Fig. 3h) with a lattice constant of 7.1 Å (indicated by the blue vectors). These two lattices were more straightforwardly illustrated in the FFT images

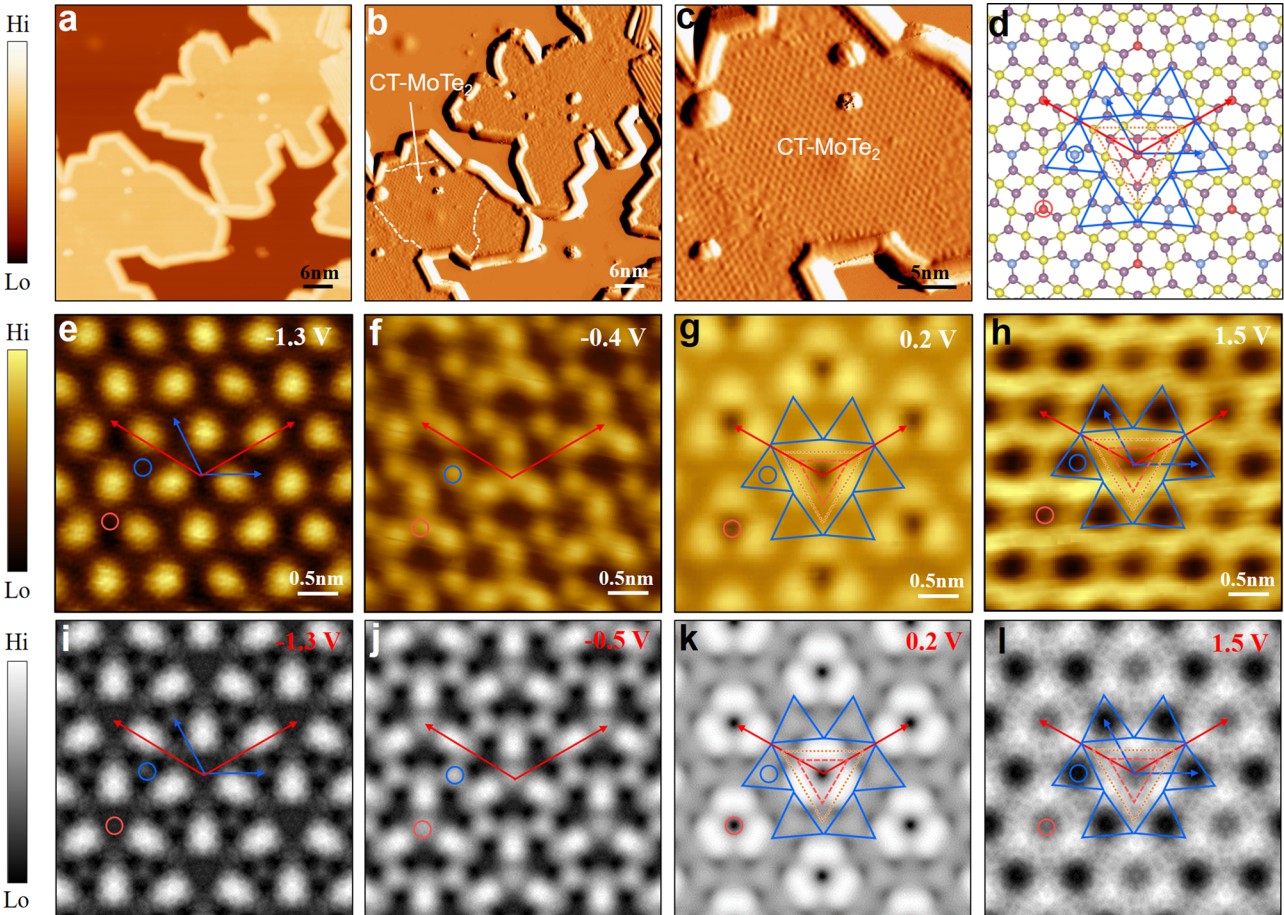

**Fig. 3 | Bias-dependent STM images of the CT-MoTe$_2$ phase. a, b** Large-scale STM topography (**a**) and current (**b**) image of the post-annealed MoTe$_2$ monolayer. White dashed lines indicate the area of the formed CT-MoTe$_2$ phase. **c** Magnified STM current image of the CT-MoTe$_2$ area. **d** Structural model of the CT-MoTe$_2$ phase. **e-h** Bias-dependent STM topography images of the CT-MoTe$_2$ phase, showing an apparent electronic Janus lattice. Generally, the primitive atomic-lattice (red, larger) and the Te pseudo-sublattice (blue, smaller) are apparently observed within and out of the energy-range of (−1V, +1 V), the lattice vectors of which were denotes using the red and blue arrows, respectively. **i-l** Simulated STM images of the CT-MoTe$_2$ phase. The red/blue and orange triangles highlight the MTTDs and MTB triangular segments of CT-MoTe$_2$. The Te$_{MTTD}$-R and Te$_{MTTD}$-B atoms are marked by the red and blue circles in (**d-l**), respectively. Scanning parameters are (**a, b**) $V = -2.3$ V, $I = -100$pA; (c) $V = -2.0$ V, $I = -100$pA.

(Supplementary Fig. 9) for those topographic ones shown in Fig. 3e–h. Lattice constants of the both lattices are consistent with the theoretical values of 12.66 and 7.31 Å of the proposed CT-MoTe$_2$ model shown in Fig. 2d. It is also subtly noted that the bright spots in Fig. 3e show a certain chiral characteristic, and one kind of pits (representing Te$_{MTTD}$-R, denoted using the red circle) appears less dark than the other (Te$_{MTTD}$-B, the blue circle) in Fig. 3h. These features, consistent with those in the simulated images (Fig. 3i and l), verify again that the prepared sample is the CT-MoTe$_2$ monolayer and indicate the electronic nature of the Te pseudo-sublattice. A similar feature was observed in the Mo$_5$Te$_8$ layer which was, however, inclusively and tentatively assigned to CDW states[39,40].

## Electronic states of CT-MoTe$_2$ phase

The experimental and theoretical consistency and visualized the appearance of the CT lattice were verified again in Fig. 4. Within an apparent DOS gap (denoted using the green shadow in Fig. 4b) comparable to that of 1H-MoTe$_2$, appreciable in-gap states were observed near the $E_F$ (Fig. 4b). A magnified tunneling spectrum of the in-gap states was acquired and plotted in Fig. 4c. It shows a clear dip at $E_F$ and two pronounced (e-S1 at 0.05 V and e-S2 at 0.28 V) and two wide peaks (e-S0 spanning from −0.35 − −0.03 V and e-S3 centered at 0.44 V) near the $E_F$. These four peaks were well reproduced in our DOS plot (Fig. 4d) as peaks t-S1 (0.05 eV), t-S2 (0.25 eV), t-S0 (from −0.37 − −0.02 eV), and

t-S3 (centered at 0.43 eV). Peak t-S1 originates from state CT2-A, and state t-S2 is contributed from three less dispersive bands of set CT3 (Supplementary Fig. 10). Wide states t-S0 and t-S3 represent the dispersive bands of breathing kagome-like CT1-B (Fig. 2g and j) and CT4 (Supplementary Fig. 10), respectively. These well-consistent assessments indicate the existence of CT and kagome lattices and, at least, one pronounced flat band (CT2-A) that sits only 0.05 eV away from $E_F$ and may host strong electron-electron interaction.

Spatial maps of states e-S0 to e-S3 and their corresponding theoretical maps for t-S0 to t-S3 were displayed in Fig. 4e–h and i–l, respectively. The state e-S0 is contributed by dispersive bands CT1-B, which exhibits protrusions around atoms Te$_{MTTD}$-R and -B in the experimental (Fig. 4e) and theoretical (Fig. 4i) maps. More detailed discussion is supplied in Supplementary Fig. 11. Flat band e-S1 (Fig. 4f), showing a full width at half maximum of ~80 meV, mostly records the spatial distribution of state CT2-A (Figs. 2k and 4j) which exhibits depressions around atoms Te$_{MTTD}$-R and -B, which were reproduced in the theoretical map (Fig. 4j). For state e-S2 (Fig. 4g), more pronounced charge density was found dominantly around Te$_{MTTD}$-R (the red circle) than that around Te$_{MTTD}$-B (the blue circle), well consistent with the theoretical map of t-S2 (Fig. 4k). This map exhibits a lattice of ~12.66 Å, which represents the unit cell of the atomic-lattice. The delocalized e-S3 (t-S3) is mostly distributed on the Te$_{MTB}$ atoms and both the Te$_{MTTD}$-R and -B atoms appear dark, as shown in Fig. 4h (4 l), which

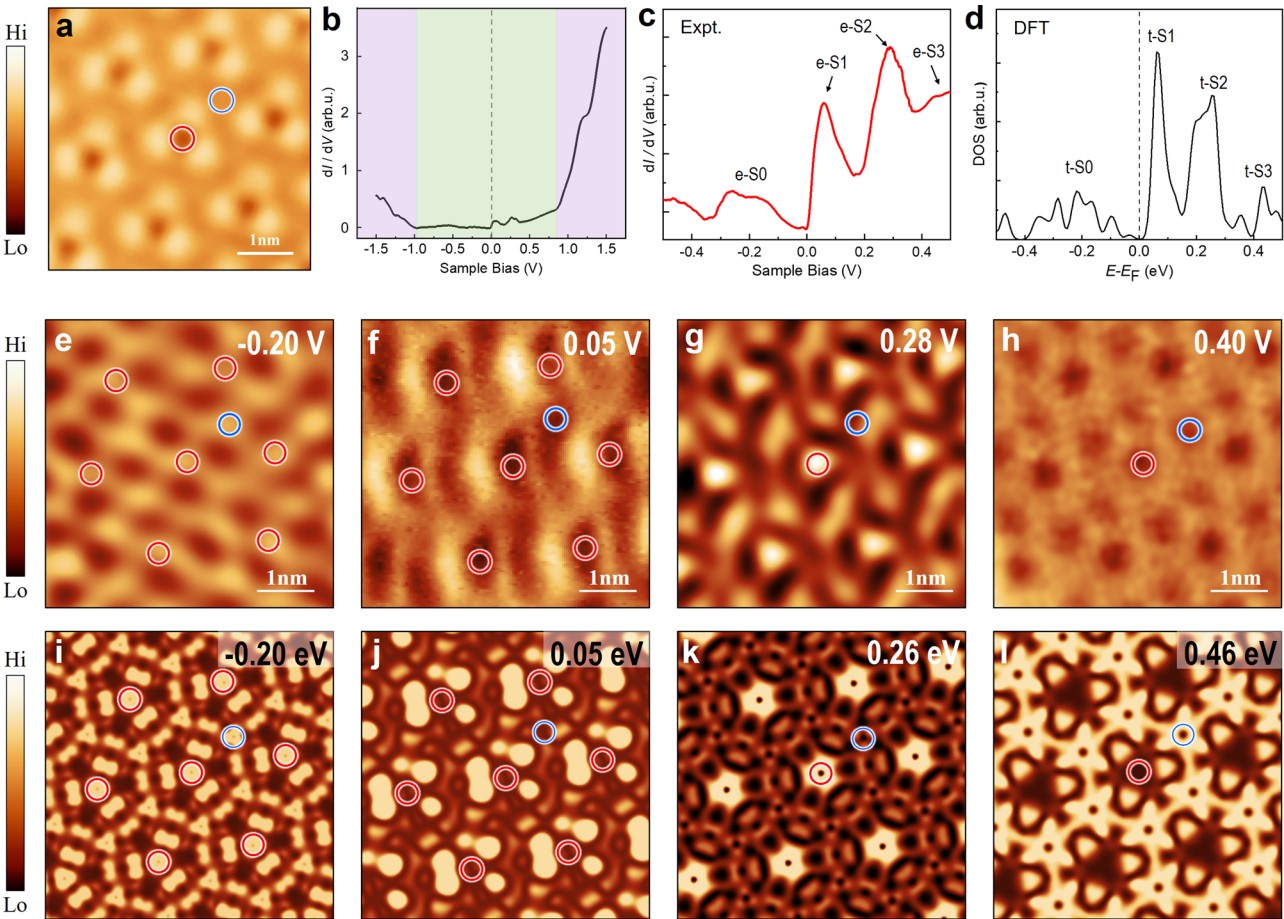

**Fig. 4 | Scanning tunneling spectroscopy (STS) measurements of the CT-MoTe$_2$ phase. a** STM topography image of the CT-MoTe$_2$ phase. **b** Large-scale averaged *dI/dV* spectrum of the CT-MoTe$_2$ phase, showing an apparent DOS gap (denoted using the green shadow) and appreciable in-gap states near the $E_F$. **c**–**d** Magnified *dI/dV* spectrum of the in-gap states (**c**) and total DOS (**d**) of the CT-MoTe$_2$ phase. Four bands of e-S0, e-S1, e-S2 and e-S3 are shown in (**c**), where e represents experimental.

**e**–**h** Constant-current *dI/dV* maps of (**a**) acquired at −0.20 V (**e**), 0.05 V (**f**), 0.28 V (**g**) and 0.40 V (**h**) (**e**–**h**, 5 nm × 5 nm), respectively, and their associated theoretically simulated maps derived from the wavefunction norms of the states sitting at −0.20 eV (**i**), 0.05 eV (**j**), 0.26 eV (**k**), and 0.46 eV (**l**) of the Γ point. The Te$_{MTTD}$-R and Te$_{MTTD}$-B atoms are marked by the red and blue circles in (**a**, **e**–**l**), respectively.

exhibits an electronic lattice of ~7.31 Å (Te pseudo-sublattice). This interesting feature of the electronic Janus lattice in the CT-MoTe$_2$ monolayer was more comprehensively demonstrated in the energy-dependent STM/STS mappings shown in Supplementary Fig. 12.

## Discussion

The CT-MoTe$_2$ monolayer, partially constituted of domain boundaries, also has domain boundaries where the electronic Janus lattice feature persists. Figure 5a and b show the STM topography images of a domain boundary acquired at different bias voltages, which exhibits an inversion symmetry of atomic structures and is thus termed the IV boundary of the CT monolayer (DB-IV). DB-IV is characterized by a ZZ-arranged Te$_{MTTD}$-B atomic chain indicated by light and dark blue triangles and one inversion center was marked using a violet cross in Fig. 5c. In STM images, DB-IV is almost indistinguishable in Fig. 5a where the image exhibits the electronic Te pseudo-sublattice, while it is explicitly imaged in Fig. 5b, in which the atomic-lattice is displayed. These images indicate that the translation symmetry of the atomic-lattice breaks in DB-IV, as plotted in the atomic structure shown in Fig. 5c, but that of the electronic Te pseudo-sublattice is nearly maintained. The energy-dependent continuality of boundary DB-IV could behave like a gate-tunable transport barrier to control flowing of charge carriers across the boundary, as more clearly illustrated in Supplementary Fig. 13. If the CT-MoTe$_2$ was integrated into an electronic device, one would expect DB-IV may promote the gating

efficiency at some certain gating voltages. Another example of domain boundaries lies in two Te$_{MTTD}$-B atomic chains forming a mirror twin (MT) boundary, thus termed DB-MT. We showed its STM images in Fig. 5d and e, while Fig. 5f displays the corresponding atomic structural model. The translation symmetry of either the atomic-lattice or the Te pseudo-sublattice is broken across this boundary, which may lead to emerging properties subject to future experimental and theoretical studies. The feature of DB-IV and DB-MT in the CT-MoTe$_2$ monolayer was more comprehensively demonstrated in the energy-dependent STM images shown in Supplementary Figs. 14 and 15.

In summary, we successfully synthesized the CT-MoTe$_2$ monolayer by introducing the highest MTB density orderly and uniformly into a pristine MoTe$_2$ monolayer using high-temperature post-growth annealing of MBE-grown MoTe$_2$ monolayers. In addition to flat electronic bands and Dirac-like states that the CT lattice symmetry inherently exhibit, the CT-MoTe$_2$ monolayer shows energy-dependent electronic Janus lattices, including the original atomic-lattice and an electronic Te pseudo-sublattice. Two types of domain boundaries were observed in the CT-MoTe$_2$ monolayer, one of which the electronic-Janus-lattice feature maintains implying application potentials in future functional devices. The atomic arrangement of CT-MoTe$_2$ inspires us to further expand the family of polymorphs in CDW phases of TMDs. A straightforward strategy lies in combining honeycomb-arranged CDW units centering an inverted CDW unit in a supercell, whose structural characteristic

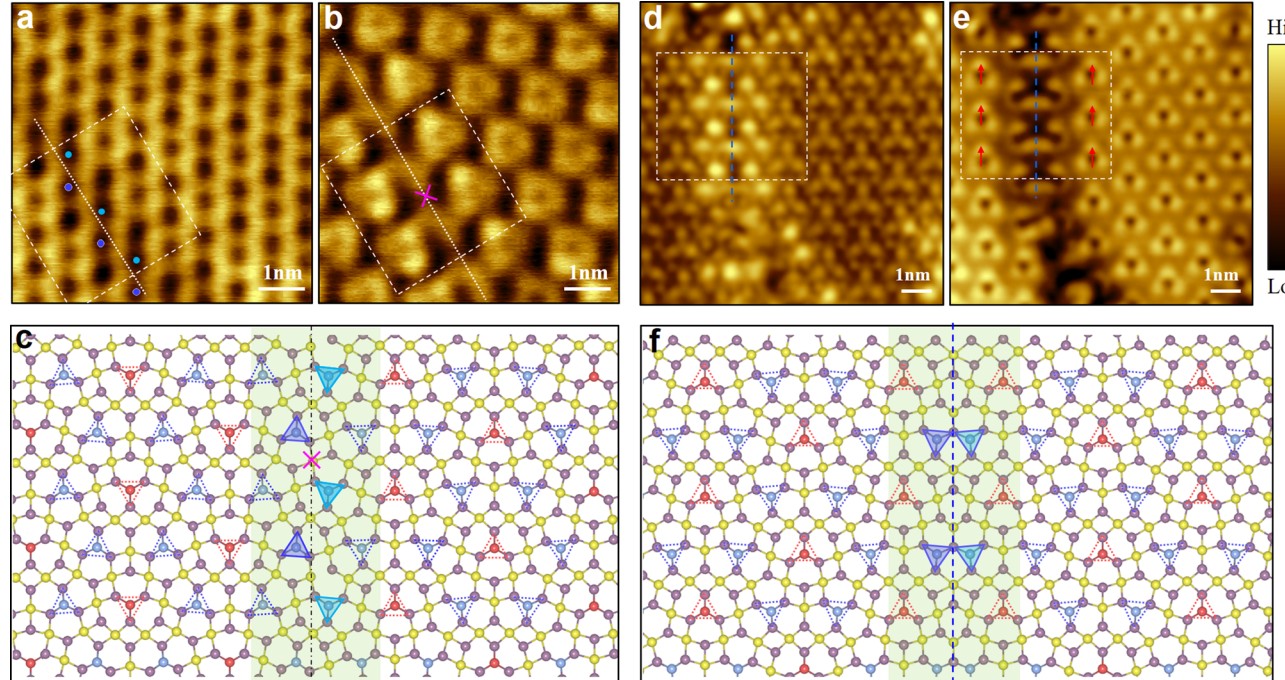

**Fig. 5 | Domain boundaries of the CT-MoTe₂ phase. a, b** STM topography images of the domain boundary with the inversion symmetry (named as DB-IV). The translation symmetry in the atomic-lattice (Te pseudo-sublattice) breaks shown by pink cross (nearly preserves shown by colored circles) across DB-IV, consistent with the found electronic Janus lattice of the CT-MoTe₂ phase. **c** Atomic structural model of DB-IV. An inversion-symmetric center at the domain boundary was marked by the pink cross. **d, e** STM topographic images of the domain boundary with the mirror twin symmetry (named as DB-MT), as the symmetry shown by the red arrows in (**e**). **f** Atomic structural model of DB-MT. Either atomic-lattice or Te pseudo-sublattice symmetry breaks across DB-MT. The dashed rectangle regions contain the domain boundaries and adjust domains, where the white dotted lines/blue dashed lines are located at the center of domain boundaries to show the symmetry in (**a**), (**b**), (**d**) and (**e**). Te$_{MTTD}$-R and -B are marked by the small red and blue dashed triangles and the green shaded areas represent the DB-IV/DB-MT region in (**c**) and (**f**). Dark- and light-blue shadowed triangles were used to denote Te$_{MTTD}$-B atoms at the boundaries to clearly demonstrate the symmetric features of DB-IV and -MT. Scanning parameters are (**a**) $V = 1.34$ V, $I = 80$pA; (**b**) $V = 1.14$ V, $I = 80$pA; (**d**) $V = -0.3$ V, $I = -100$pA; (**e**) $V = 0.3$ V, $I = 100$pA.

follows that of the CT-MoTe₂ (Supplementary Fig. 16). It was demonstrated that CDW structures in TMDs could be "condensed" selectively by point defects[21,46], indicating the above scenario highly promising. Our work offers an effective route to artificially build structural polymorphs in TMDs that host exotic electronic properties to be explored.

## Methods
### Sample preparation
The single-layer MoTe₂ films were grown on highly oriented pyrolytic graphite (HOPG) substrate in a home-built MBE system with a base pressure of $3.0 \times 10^{-10}$ Torr. The highly oriented pyrolytic graphite (HOPG) substrate was freshly cleaved in air and immediately loaded into the ultra-high vacuum (UHV) chamber of MBE, then degassed at 773 K overnight to remove contaminants. The high-purity Mo (99.999%) and Te (99.999%) were simultaneously evaporated from an electron beam evaporator and a Knudsen cell, respectively. The temperature of HOPG during growth was ~513 K. After growth, all samples were followed by annealing with either a growth temperature maintained or higher temperature (616 K). The sample was monitored by beam flux monitor (BFM) and reflection high-energy electron diffraction (RHEED) to regulate the temperature of both the source and annealing process to form CT-MoTe₂ phase.

### STM measurements
The samples were transferred to another UHV chamber with LT-STM (PanScan Freedom, RHK) for the following STM measurements. All STM/STS measurements were performed at 9 K with a chemically etched W tip calibrated on a clean Ag(111) surface (Supplementary Fig. 17). The STM images were acquired in constant-current mode. The

$dI/dV$ spectra were obtained by using a standard lock-in amplifier with bias modulation ~5 mV at 857 Hz. All STM images were processed by Gwyddion and WSxM[47] software.

### DFT calculations
Density functional theory calculations were performed using the generalized gradient approximation for the exchange-correlation potential, the projector augmented wave method, and a plane-wave basis set as implemented in the Vienna Ab initio Simulation Package (VASP)[48]. The energy cutoff for plane wave was set to 500 eV for invariant volume structural relaxation of freestanding CT-MoTe₂ monolayers. A dispersion correction was made at the van der Waals density functional (vdW-DF) level, with the optB86b functional for the exchange potential[49]. During all structural relaxations, all atoms were fully relaxed until the residual force per atom was less than $1 \times 10^{-2}$ eV Å$^{-1}$ and the energy convergence criteria was $1 \times 10^{-5}$ eV. The lattice constant is 12.66 Å after relaxation. The isosurface values for theoretically simulated $dI/dV$ maps are $1 \times 10^{-2}$ e Bohr$^{-3}$ and $1 \times 10^{-3}$ e Bohr$^{-3}$. A $7 \times 7 \times 1$ k-mesh was used to sample the first Brillouin zone in all calculations. An effective on-site Coulomb energy $U = 1.5$ eV was considered in all calculations. The thickness of the vacuum layer is set to 15 Å. In plotting DOS spectra, a Gaussian smearing of 0.04 eV was used. The energy level of $E_F$ was set to energy zero in DOS and band structure calculations.

## Data availability
Relevant data supporting the key findings of this study are available within the article and the Supplementary Information file. All raw data generated during the current study are available from the corresponding authors upon request.

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

## Acknowledgements

This project is supported by Strategic Priority Research Program and Key Research Program of Frontier Sciences and Instrument Developing Project (Chinese Academy of Sciences, CAS) [No. XDB30000000 (Z.C. and W.J.), No. QYZDB-SSW-SYSO31 (Z.C.), No. YZ201418 (Z.C.)], the National Natural Science Foundation of China (NSFC) [No. 21622304 (Z.C.), 61674045 (Z.C.), 11604063 (R.X.), 11974422 (W.J.), 12104504 (W.J.)], the National Key R&D Program of China [Grant No. 2018YFE0202700 (Z.C.)]. Z. H. Cheng was supported by Distinguished

Technical Talents Project and Youth Innovation Promotion Association CAS, the Fundamental Research Funds for the Central Universities and the Research Funds of Renmin University of China [No. 21XNLG27 (Z.C.), No. 22XNKJ30 (W.J.), No. 22XNH095 (H.D.)]. Calculations were performed at the Physics Lab of High-Performance Computing of Renmin University of China, Shanghai Supercomputer Center and Beijing Supercomputing Center.

## Author contributions

Z.C., G.W. and W.J. conceived the research project. L.L., H.D., Y.G. and Z.C. performed the STM experiments and analysis of STM data. R.X., F.P. and F.L. helped in the experiments. J.D., F.C., C.W., Z.L., and W.J. performed the DFT calculations. L.L., J.D., F.C., Z.C. and W.J. wrote the manuscript with inputs from all authors.

## Competing interests

The authors declare no competing interests.
