## [Peer Review File · Nature Communications]

Electronic Janus lattice and kagome-like bands in coloring-triangular MoTe₂ monolayersEditorial Note: Parts of this Peer Review File have been redacted as indicated to remove third-party material where no permission to publish could be obtained.

REVIEWER COMMENTS

Reviewer #1 (Remarks to the Author):

This paper reports a combined experimental and theoretical study of coloring-triangle (CT) lattice of MoTe₂, which is synthesized by design via MBE and characterized in depth using STM/STS and DFT. Overall, I think the approach they took is innovative and new and the work is very solid. Therefore, I essentially recommend publication of this work after the authors address the following questions: (1) One deficiency of this work is the explanation of band structure, especially the flat band of CT lattice. It is noted that physically flat band arises purely from lattice symmetry of a Kagome or CT lattice, assuming an s or p_z orbital (even parity) on each lattice site (see, e.g., H. Liu et al., PRB 105, 085128 (2022)). However, their orbital-resolved DFT bands showed the FB is composed of p- or d-orbitals. If one simply uses default p- (such as p_x and p_y they mentioned) or d-orbital on each CT lattice site, I would expect no flat band (the authors may test and confirm this using tight-binding model calculation). Therefore, it will be helpful if authors can dig deeper into this aspect to further strengthen their work.

(2) The concept of “electronic Janus lattice” is not clear, at least to general readers. Usually the Janus lattice refers to mirror-asymmetric 2D layers with out-of-plane polarization. They seemed to be talking something different?

(3) MTB is usually metallic due to symmetry, as they mentioned in the introduction. Figure 1f showed a gapped MTB electronic state, which is not explained. Have they constructed a supercell with MTB to calculate the band structure?

Reviewer #2 (Remarks to the Author):

The authors report a new structural and electronic phase of the transition metal dichalcogenide MoTe₂. This phase consists of a hexagonal arrangement of the smallest Mo-terminated triangular domains, and was enabled by the high-density periodic introduction of mirror twin boundaries. Interestingly, density functional theory (DFT) calculations (with a nonzero Hubbard U accounting for electron-electron interactions) show that the band structure of such a MoTe₂ phase includes Dirac bands and two quasi-flat bands near the Fermi level.

The study is timely, given the current emphasis in materials research on 2D materials with flat bands (i.e., with potential to host strong electronic correlations and diverse many-body quantum phases). The experimental and theoretical results are of good quality, most often supporting the authors’

claims.

In my opinion, the manuscript deserves to be published in Nature Communications, provided that the following comments are addressed and taken into account in a following version of the draft:

- The authors insist that the electronic structure of the CT-MoTe₂ system includes two flat bands, one at ~0.05 eV and the other at ~0.20-0.25 eV. The latter is more 'quasi-flat' than perfectly flat. The authors should be specific on this.

- In Fig. 4, the claimed resemblance between experimental dI/dV maps (e-g) and corresponding theoretical maps (h-j) is not that convincing. In particular, states associated with features S1 and S2 (i.e., quasi-flat bands) should be strongly localised, which is the case in the theoretical maps, but not at all obvious in the experimental ones. Moreover, the magnitude of these S1 and S2 dI/dV features in Fig. 4a and c is very small in comparison to the valence band maximum (VBM) and conduction band minimum (CBM) states in Fig. 4a. This could seem surprising, given that S1 and S2 are related to flat bands (i.e., sharp DOS resonances at the corresponding energies). The authors should comment on this. How do the theoretical features t-S0, t-S1 and t-S2 compare to the theoretical VBM and CBM? It would be good for the authors to show a theoretical DOS similar to Fig. 4d, but from -1.5 to 2 eV, including the VBM and CBM. Also, how do an experimental dI/dV map and corresponding theoretical map compare at an energy related to S0 (e.g., at ~-0.2 eV)? This is also related to the claim in lines 253-254 on p. 11: "Highly localized charge density...(red circles in Fig. 4f)", with which I disagree; other features not circled in red show similar magnitudes in Fig. 4f. The authors should provide a comment and explanation for this.

- Lines 273-274, p. 11: The authors should elaborate on the claim "Boundary DB-IV, separating two adjacent domains. ... barrier resisting carrier flowing". I don't really understand what is meant by this.

Minor comment:

- I understand that in the DFT calculations U was fixed such as to reproduce the MoTe₂ bandgap. It would be good if the authors could justify this value of U, e.g., is it the results of the intrinsic Mo d states? Or of the lattice geometry of the MoTe₂ polymorph, which, via the quasi-flat bands, could give rise to significant electronic correlations? Are similar values of U used in prior DFT studies of MoTe₂?

Reviewer #3 (Remarks to the Author):

Referee report:

Manuscript "Electronic Janus lattice and Kagome-like bands in coloring-triangular MoTe₂ monolayers" by Lei and coworker reports a theoretical design and experimental realization of a coloring-triangular (CT) lattice in Te deficient MoTe₂ monolayer, in particular, a Mo₅Te₈ monolayer. This work is the first indication that the Mo₅Te₈ monolayer is a CT lattice, which exhibits a novel

electronic Janus lattice and several sets of kagome-like bands. The theoretical and experimental results are of very high quality and well consistent, and the found CT-MoTe₂ exhibiting interesting Janus lattices and kagome bands is indeed intriguing. To the best of my knowledge, this work is also the first realization of the CT lattice in 2D monolayers and, perhaps, in all crystals. It also demonstrated the ability of tuning intralayer domain size and boundary, rather than interlayer twisting and moiré potential in hetero-bilayers, to engineer properties of TMDs. Overall, this is an exciting piece of work in the field of TMD materials and I could see it large impact in the 2D and kagome communities. Before I could recommend it for publication in Nature Communications, there are several points that the authors should clarify.

- (1) The authors mentioned “this series of images shows two apparent lattices” in Fig. 3e-3h. Is it possible to distinguish these two lattices in the FFT image of STM topography images or dI/dV conductance maps?
- (2) The pseudo-lattice is clearly observable under high bias voltages, i.e. below -1 V or above 1 V. Does it have any intrinsic reason? If the pseudo-lattice dominates in a wide range of energy, it would be curious to know whether the electronic bandstructures follow the Brillouin zone of the atomic lattice or the pseudo-lattice?
- (3) It would be interesting to briefly explain why the domain boundary displayed in Fig. 5a is invisible at certain bias voltages. Is it relevant to a unique electronic state? Does it have application potentials in further electronic devices?
- (4) Kagome lattice is a well-known lattice, but CT is recently proposed one which is not as familiar as the kagome lattice to the 2D community. Could the authors comment on the connection(s) between the kagome and CT lattices?
- (5) As mentioned in the methods section, the CT-MoTe₂ monolayer was prepared by controllably desorption of Te atoms from the MoTe₂ monolayer. Could the authors clarify the route(s) to monitor and control the desorption of Te accurately?
- (6) If the Te desorption process is precisely controllable, could one get a monolayer comprised of those N=3 or N=6 uniform MTB triangles? If yes, which structures are energetically more favored? Are they experimentally accessible?
- (7) The post-growth annealing method aside, are there any other likely routes to synthesize this CT-MoTe₂ monolayer? If yes, could these methods applicable to other TMD materials like MoSe₂, NbTe₂, TaTe₂?

There are some minor technical issues, see comments below:

- (1) There are three labels, namely 1T', 1H and MTB, marked in Fig.1b. I wonder if the domain labeled using MTB appropriate or should it be presented by another label?
- (2) Term “TeMTTD-R” was used to identify a certain group of Te atoms. Could the authors provide a brief explanation to this term?
- (3) In the figure caption of Fig.4, it says “...The TeMTTD-R and TeMTTD-B atoms are marked by the red and blue circles in (b, d-l), respectively...”. Here, panel “d-l” should, perhaps, read as panel “e-j”.

REVIEWER COMMENTS

Reviewer #1 (Remarks to the Author):

This paper reports a combined experimental and theoretical study of coloring-triangle (CT) lattice of MoTe_2 , which is synthesized by design via MBE and characterized in depth using STM/STS and DFT. Overall, I think the approach they took is innovative and new and the work is very solid. Therefore, I essentially recommend publication of this work after the authors address the following questions:

We thank the Reviewer for his/her careful review, constructive comments, and the positive recommendation.

Comment 1-1: *One deficiency of this work is the explanation of band structure, especially the flat band of CT lattice. It is noted that physically flat band arises purely from lattice symmetry of a Kagome or CT lattice, assuming an s or p_z orbital (even parity) on each lattice site (see, e.g., H. Liu et al., PRB 105, 085128 (2022)). However, their orbital-resolved DFT bands showed the FB is composed of p - or d -orbitals. If one simply uses default p - (such as p_x and p_y they mentioned) or d -orbital on each CT lattice site, I would expect no flat band (the authors may test and confirm this using tight-binding model calculation). Therefore, it will be helpful if authors can dig deeper into this aspect to further strengthen their work.*

Reply 1-1:

We thank the Reviewer for this enlightening comment. We agree with the Reviewer that introduction of two- or even four-fold symmetric orbitals occupying CT-latticed sites may disturb the kagome band structures. We thus carried out additional calculations to exhaustively respond this concern and the Reviewer's request on providing more information on the theory part.

Figure R1-1 (Supplementary Fig. 5) (a) Schematic plot of CT lattice composed of Mo atoms. Orbitals are shown as filled circles and marked with red numbers, while hopping terms are shown as solid red lines. Thick black arrows denote the lattice vectors. (b-c) Band structure of CT lattice model, in which each atom carries one p_z (b), p_x (c), or p_y (d) orbital.

We first examined the Reviewer’s expectation that using p_x or p_y orbitals to occupy each CT site. Figure R1-1a (Supplementary Fig. 5a) shows the CT lattice and all hopping paths in a supercell using the TBPLaS code [*Computer Physics Communications* **285**, 108632 (2023)]. As expected by the Reviewer, the p_z orbital case exhibits a perfect set of kagome band structures (Fig. R1-1b). However, the perfect kagome band structures degrade in the p_x or p_y case (Fig. R1-1c and R1-1d) because the overlapping matrixes vary among different sites. These results thus raise a critical issue that whether the “orbitals”, of those “superatoms”, namely the Mo- and Te-terminated triangular domains (Mo₆Te₂, MTTD and Mo₆Te₂₀, TTTD), used in the CT-MoTe₂ monolayer have nonidentical overlapping matrixes?

Figure R1-2 (Supplementary Fig. 6) (a-d) Atomic structures of (a) MTTD (highlighted by red dashed triangles), (b) TTTD (highlighted by orange dashed triangles), (c) MTB surrounding structures, referred to as frame and (d) Mo₅Te₈ (CT-MoTe₂) monolayer. (e-g) Isosurface contours of wavefunction norms for Mo₅Te₈, TTTD and MTTD, and MTB frame. (e) CT1-A state of Mo₅Te₈ at Γ point using an isosurface of 1×10^{-3} e Bohr⁻³, mainly contributed by the d_{z^2} orbitals of Mo atoms on vertices of CT lattice and triangle domain. (f) show the calculated real-space distributions for CT1-A of TTTD and MTTD at Γ point. Isosurface is 5×10^{-3} e Bohr⁻³. (g) show the distributions of MTB frame. Three CT sites in a primitive cell were highlighted using red filled circles. (h)-(j) The same scheme of plots as that of panels (f) for CT2-B, CT3-A, and CT4-A, respectively.

Figure R1-2a (Supplementary Fig. 6a) and R1-2b show the atomic structures of MTTD and TTTD. Their relations with that of the CT monolayer were illustrated in Fig. R1-2c and R1-2d. Figure R1-2f and R1-2h to R1-2j depict their frontier orbitals, which are involved in forming bands CT1 to CT4. Most of these frontier orbitals are comprised of Mo d and Te p

orbitals, including the out-of-plane z component and the C_3 symmetrized in-plane component, providing identical overlapping matrixes among different sites. In addition, Fig. R1-2g clearly shows that the d_{z^2} orbital dominates each CT sites for the MTB framework. In light of both results, unlike those atomic p_x or p_y orbitals, introduction of these C_3 symmetrized orbitals for MTTD and/or TTTD does not change the original symmetry of the CT lattice, which leads to the kagome bands intrinsically originated from the symmetry of the CT lattice. We believe these results answer the Reviewer's concerns on the origin of the kagome band structures.

Figure R1-3 (revised Fig. 2e). Calculated band structure near the Fermi level (-1 eV to 1 eV). Four sets of CT band (CT1, CT2, CT3, and CT4) are marked with green, blue, pink, and grey lines, respectively. The irreps of states at Γ or K points are labelled for these twelve bands.

In order to provide even more information on the theory part, we next analyzed the bandstructures of the CT-MoTe₂ monolayer using the group theory based on our results from additional band-structure calculations using the Wannier90 code [*J. Phys. Cond. Matt.* **32**, 165902 (2020)] in conjunction with VASP. We, at least, identified four sets of kagome bands (Figure R1-3) based on symmetry analysis listed in Table R1-1.

The atomic structure of CT-MoTe₂ (Mo₅Te₈) belongs to space group (SG) No. 189 and exhibits the D_{3h} point group which is the direct product of point group D_3 and a mirror symmetry operation σ_h . The character table of the Γ/K -little group in SG 189 was used to determine all irreducible representations (IRs) for the CT-bands (Table R1-1). We used the Γ point for illustration, Γ_1 to Γ_6 represent six irreducible representations of D_{3h} : A'_1-E'' , respectively; operations 1 to 12 represent twelve symmetry operations: E , $2C_3$, σ_h , $2S_3$, $3C_2$, and $3\sigma_v$. The IRs of four sets of the CT bands (CT1 to CT4, see Fig. R1-3) at the Γ point were found to be (Γ_1, Γ_5) , (Γ_6, Γ_2) , (Γ_3, Γ_6) , and (Γ_5, Γ_1) , respectively. Sets CT1/CT4

and CT2/CT3 are connected through a mirror symmetry operation σ_h that $\Gamma 1$ and $\Gamma 5$ of bands CT1/CT4 have character 1, while $\Gamma 2, \Gamma 3$ and $\Gamma 6$ have character -1 for bands CT2/CT3. A similar mirror symmetry connection was observable in bands CT1 to CT4 at the K point.

Γ/K	1	2	3	4	5	6	7	8	9	10	11	12
$\Gamma 1/K1$	1.00+0.00i	1.00+0.00i	1.00+0.00i	1.00+0.00i	1.00+0.00i	1.00+0.00i	1.00+0.00i	1.00+0.00i	1.00+0.00i	1.00+0.00i	1.00+0.00i	1.00+0.00i
$\Gamma 2/K2$	1.00+0.00i	1.00+0.00i	1.00+0.00i	-1.00+0.00i	-1.00+0.00i	-1.00+0.00i	1.00+0.00i	1.00+0.00i	1.00+0.00i	-1.00+0.00i	-1.00+0.00i	-1.00+0.00i
$\Gamma 3/K3$	1.00+0.00i	1.00+0.00i	1.00+0.00i	-1.00+0.00i	-1.00+0.00i	-1.00+0.00i	-1.00+0.00i	-1.00+0.00i	-1.00+0.00i	1.00+0.00i	1.00+0.00i	1.00+0.00i
$\Gamma 4/K4$	1.00+0.00i	1.00+0.00i	1.00+0.00i	1.00+0.00i	1.00+0.00i	1.00+0.00i	-1.00+0.00i	-1.00+0.00i	-1.00+0.00i	-1.00+0.00i	-1.00+0.00i	-1.00+0.00i
$\Gamma 5/K5$	2.00+0.00i	-1.00+0.00i	-1.00+0.00i	2.00+0.00i	-1.00+0.00i	-1.00+0.00i	0.00+0.00i	0.00+0.00i	0.00+0.00i	0.00+0.00i	0.00+0.00i	0.00+0.00i
$\Gamma 6/K6$	2.00+0.00i	-1.00+0.00i	-1.00+0.00i	-2.00+0.00i	1.00+0.00i	1.00+0.00i	0.00+0.00i	0.00+0.00i	0.00+0.00i	0.00+0.00i	0.00+0.00i	0.00+0.00i
$\Gamma 7/K7$	2.00+0.00i	-2.00+0.00i	-2.00+0.00i	0.00+0.00i	0.00+0.00i	0.00+0.00i	0.00+0.00i	0.00+0.00i	0.00+0.00i	0.00+0.00i	0.00+0.00i	0.00+0.00i
$\Gamma 8/K8$	2.00+0.00i	1.00+0.00i	1.00+0.00i	0.00+0.00i	-1.73+0.00i	-1.73+0.00i	0.00+0.00i	0.00+0.00i	0.00+0.00i	0.00+0.00i	0.00+0.00i	0.00+0.00i
$\Gamma 9/K9$	2.00+0.00i	1.00+0.00i	1.00+0.00i	0.00+0.00i	1.73+0.00i	1.73+0.00i	0.00+0.00i	0.00+0.00i	0.00+0.00i	0.00+0.00i	0.00+0.00i	0.00+0.00i

$D_{3h}(-6m2)$	1	2	3	4	5	6	SYMMETRY OPERATIONS					
Mult.	A_1'	A_1''	A_2'	A_2''	E'	E''	E	2 C_3	σ_h	2 S_3	3 C_2	3 σ_v

Table R1-1 (Supplementary Table I). Character table of Γ/K -little group in space group 189. The six irreducible representations of D_{3h} and twelve symmetry operations are show in the bottom row of the table.

Action 1-1:

1) We put Fig. R1-1 and R1-2, and Table R1-1 and their associated discussions into the SI as Supplementary Fig. 5 and 6 and Table I, respectively. We also replaced Fig. 2e with Fig. R1-3.

2) We added the sentences of “The bands of sets CT1 (green) ... Fig. 5 and 6.” in page 7 and the paragraph “An interesting question is thus ... in the present work.” In page 8.

3) We added a new author, Dr. Zheng-Xin Liu for his contribution on the symmetry analysis as discussed here.

Comment 1-2: *The concept of “electronic Janus lattice” is not clear, at least to general readers. Usually the Janus lattice refers to mirror-asymmetric 2D layers with out-of-plane polarization. They seemed to be talking something different?*

Reply 1-2:

We thank the Reviewer for raising this important question. We analogized the observed electronic structures in CT-MoTe₂ to the atomic characteristics of Janus monolayers. As shared with the Reviewer, a Janus monolayer appears like two kinds of monolayers when looking from the top- and bottom-views, in which term “Janus” refers to atomic structures. Here, the electronic states of the CT-MoTe₂ monolayer exhibit two (quasi-)periodicities in their real-space distributions, representing the atomic-lattice and the Te pseudo-sublattice, each of which could be observed at certain energies (bias voltages in STM images). In light

of this, the electronic characteristics of CT-MoTe₂ are “double-faced”. Thus, we term this feature as the electronic Janus lattice, in which term “Janus” focuses on electronic structures.

Figure R1-4. Comparison of Structural Janus lattice (MoSSe) and Electronic Janus lattice (CT-MoTe₂). (a) Top and side views of monolayer MoSSe. (b-e) Bias-dependent STM topography images of the CT-MoTe₂ phase, showing an apparent electronic Janus lattice. Generally, the primitive atomic-lattice and the Te pseudo-sublattice are apparently observed on low bias voltages and high bias voltages, the lattice vectors of which were denotes using the red and blue arrows, respectively.

Action 1-2:

1) We explained the meaning of electronic Janus lattice and analogized/distinguished it with/from the Janus monolayer.

2) We revised the sentences of “The both pronounced electronic contributions...in our following experiments.” to “Both pronounced electronic contributions ... focuses on electronic structures.” in the first paragraph on page 8.

Comment 1-3: MTB is usually metallic due to symmetry, as they mentioned in the introduction. Figure 1f showed a gapped MTB electronic state, which is not explained. Have they constructed a supercell with MTB to calculate the band structure?

Reply 1-3:

We thank the Referee for raising this important question. An infinite MTB in MoS₂ indeed exhibits an 1D metallic state, as shown in Fig. R1-5 [*Phys. Rev. X* **9**, 011055, (2019)], which is shared in MTBs of MoSe₂ [*Nat. Phys.* **12**, 751-756 (2016)] and MoTe₂ [*ACS Nano* **14**, 8299-8306 (2020)] monolayers. That is the reason why we stated “These MTBs are metallic and show a high density of states (DOS) near the Fermi level” in the introduction. We regret that this statement is a bit inaccurate. We shall emphasize that a pristine infinite-length MTB is theoretically revealed to be metallic, however, there are, at least, two reasons

that the measured MTBs show a gapped feature in dI/dV spectra in our samples.

Figure Redacted

For a finite-length MTB, as we measured in the present work, the 1D metallic state may open a gap because of quantum confinement along its propagating direction. In addition, such 1D metallic state crosses the Fermi level, which is less stable and may undergo transitions to open a gap further stabilizing the Fermi surface at low temperatures. Such transitions were observed in the literature that MTBs in MoSe_2 and MoTe_2 , even in finite lengths, undergo Peierls phase transitions forming charge density wave (CDW) gaps around the Fermi level [see, Fig. R1-6, *Phys. Rev. X* **9**, 011055, (2019), *Nat. Phys.* **12**, 751-756 (2016)], in which further charge-spin separation was found in MoSe_2 to suggest a Tomonaga-Luttinger liquid state [*Nat. Mater.* **21**, 748-753 (2022)]. Plots of STS acquired at MTB of MoS_2 (Fig. R1-6a) and MoTe_2 (Fig. R1-6b) both exhibit narrow gaps of approximately 100 meV residing around the Fermi level, which are consistent with our results shown in Fig. R1-6c.

Figure Redacted

Action 1-3:

We revised the sentences of “These MTBs are metallic ... like Peierls-type CDW^{31, 32} or Tomonaga-Luttinger liquid^{33, 34} at low temperature.” in the second paragraph on page 2 to “Infinite-length MTBs were theoretically revealed to be metallic ... like Peierls-type CDW^{31,32} or Tomonaga-Luttinger liquid^{33,34}”.

Reviewer #2 (Remarks to the Author):

The authors report a new structural and electronic phase of the transition metal dichalcogenide MoTe_2 . This phase consists of a hexagonal arrangement of the smallest Mo-terminated triangular domains, and was enabled by the high-density periodic introduction of mirror twin boundaries. Interestingly, density functional theory (DFT) calculations (with a nonzero Hubbard U accounting for electron-electron interactions) show that the band structure of such a MoTe_2 phase includes Dirac bands and two quasi-flat bands near the Fermi level.

The study is timely, given the current emphasis in materials research on 2D materials with flat bands (i.e., with potential to host strong electronic correlations and diverse many-body quantum phases). The experimental and theoretical results are of good quality, most often supporting the authors' claims.

In my opinion, the manuscript deserves to be published in Nature Communications, provided that the following comments are addressed and taken into account in a following version of the draft:

We thank the Reviewer for his/her careful review, constructive comments, and the positive recommendation.

Comment 2-1.

The authors insist that the electronic structure of the CT- MoTe_2 system includes two flat bands, one at ~ 0.05 eV and the other at ~ 0.20 - 0.25 eV. The latter is more 'quasi-flat' than perfectly flat. The authors should be specific on this.

Reply 2-1:

We are grateful to the Referee for this constructive comment. We analyzed more bands shown in the revised Fig. 2e. The STS/DOS peak sitting at ~ 0.20 - 0.25 eV belong to bands CT3. It is a set of kagome bands according to our symmetry analysis shown in Supplementary Table I. However, its band-width is only 0.1 eV and the flat band of CT3 mixes up with the two Dirac bands of it, leading the appearance of t-S2 (e-S2) broader than that of t-S1 (e-S1, CT2). We agree with Reviewer that the pronounced peak found at ~ 0.20 - 0.25 eV in STS and DOS plots is not ascribed to a flat band solely. We revised the descriptions of this peak in the revision.

Action 2-1:

1) We deleted the sentence of "Two sets of flat-bands reside at approximately 0.05 and 0.20 eV (over E_F , hereinafter)" in the first paragraph on page 7 and added the sentences of "The flat band of CT2...highly dispersive." in the second paragraph on page 7.

2) We revised the sentence of "... , at least, two sets of their associated flat bands..." in the first paragraph on page 7 to "... , at least, one pronounced flat band (CT2-A) ...".

Comment 2-2-(1): In Fig. 4, the claimed resemblance between experimental dI/dV maps (e-g) and corresponding theoretical maps (h-j) is not that convincing. In particular, states associated with features S1 and S2 (i.e., quasi-flat bands) should be strongly localized, which is the case in the theoretical maps, but not at all obvious in the experimental ones.

Reply 2-2-(1):

We thank the Review for raising this issue. We regret with such misleading made to the Reviewer. These theoretical maps shown in Fig. 4h-4j were intended to show extensiveness of e-S1 and e-S2, not localization, which distinguishes flat bands resulted from spatial isolation (localized, flat atomic bands, FAB) and destructive interference (extended, flat topological bands, FTB) of electron wavefunctions. The flat bands discussed in our work are the extended FTB (flat topological bands) with completely extended wave functions. The detailed discussion about the two kinds (FTB vs FAB) of flat bands can be found in the literature (*Catalogue of flat-band stoichiometric materials. Nature* **603**, 824-828 (2022)). The extensiveness of the flat bands (FTB) in kagome lattices was also shared in the literatures, e.g. *Sci. Adv.* **4**, eaau4511 (2018) and *ACS. Nano.* **16**, 21079-21086 (2022).

Action 2-2-(1):

We replotted original Fig. 4e-4j by revising its colormap and color scalebar and added two panels (see Reply2-2-(3)). The results of theoretically simulated maps are more consistent with experimental maps in the revision.

Figure R2-1 (revised Fig. 4e-l) (a-d) Constant-current dI/dV maps of CT-MoTe₂ acquired at -0.20 V (a), 0.05 V (b), 0.28 V (c) and 0.40 V (d), respectively, and their associated theoretically simulated maps derived from the wavefunction norms of the states sitting at -0.20 V (e), 0.05 eV (f), 0.26 eV (g), and 0.46 eV (h) of the Γ point. The Te_{MTTD-R} and Te_{MTTD-B} atoms are marked by the red and blue circles in (a-h), respectively.

Comment 2-2-(2) Moreover, the magnitude of these S1 and S2 dI/dV features in Fig. 4a and c is very small in comparison to the valence band maximum (VBM) and conduction band minimum (CBM) states in Fig. 4a. This could seem surprising, given that S1 and S2 are related to flat bands (i.e., sharp DOS resonances at the corresponding energies). The authors should comment on this. How do the theoretical features t-S0, t-S1 and t-S2 compare to the theoretical VBM and CBM? It would be good for the authors to show a theoretical DOS similar to Fig. 4d, but from -1.5 to 2 eV, including the VBM and CBM.

Reply 2-2-(2):

We thank the Reviewer for raising this interesting question. We plotted the theoretical DOS of the CT-MoTe₂ phase from -1.5 to 1.5 eV, as the Reviewer suggested, in Fig. R2-2a, in which the magnitudes of states t-S1 and t-S2 are comparable to those of the VB and CB states. However, we have to emphasize that it is not feasible to directly compare measured dI/dV spectra with theoretical DOSs (semi-)quantitatively, as we elucidated as follows.

A dI/dV spectral measurement represents conductance of the tip-vacuum-sample junction, which is the convolution between the DOSs of the sample and the tip, weighted by the transmission coefficients of the vacuum barrier that, at least, depends exponentially on bias voltage. Thus, it requires normalizations of the raw dI/dV spectra acquired at finite bias voltages for making semi-quantitative comparisons with theoretical DOSs. A popularly used normalization method lies in $(dI/dV)/(IV)$ [*Phys. Rev. Lett.* **57**, 2579 (1986)]. Figure R2-2b shows the normalized conductance spectrum, namely $(dI/dV)/(IV)$, of the CT-MoTe₂ phase, in which the magnitudes of states e-S1 and e-S2 are largely enhanced and better comparable with those of states t-S1 and t-S2 appeared in the theoretical DOS.

Figure R2-2 STS measurements of the CT-MoTe₂ phase. (a) The theoretically simulated total DOS of CT-MoTe₂ phase from -1.5 to 1.5 V. (b) The large-scale normalized differential conductance $(dI/dV)/(IV)$ spectrum of CT-MoTe₂ phase. The normalized differential conductance $(dI/dV)/(IV)$ spectrum is relatively consistent with the calculated DOS of CT-MoTe₂.

Comment 2-2-(3) Also, how do an experimental dI/dV map and corresponding theoretical map compare at an energy related to $S0$ (e.g., at ~ -0.2 eV)?

Reply 2-2-(3):

We thank the Reviewer for raising this question and added Supplementary Fig. 11 (Fig. R2-3) in the revision. State e-S0 ranges from -0.35 to -0.03 V in the experimental spectrum (Fig. R2-3a), which corresponds to bands CT1-B1 and CT1-B2 in our DFT calculations (Fig. R2-3c). Bands CT1-B1 and CT1-B2 are ascribed to a Dirac band and a nominal “flat band”, respectively. Term “nominal” means that CT1-B2 is different from a usual flat band, which contacts CT1-B1 at the K point and is highly distorted and more dispersive.

We showed a dI/dV map acquired at -0.20 V and its associated theoretical image obtained at -0.20 ± 0.02 eV in Fig. R2-3d and e. The dI/dV map exhibits two patterns, namely a triangular one (marked by blue triangles) being embedded in a distorted hexagonal one (marked by green hexagons), which are roughly consistent with the corresponding theoretical map. The hexagonal and embedding triangular patterns are also consistent with their origins from a Dirac state (CT1-B1) and a nominal “flat band” (CT1-B2). Because CT1-B2 is much more dispersive, the appearance of its associated DOS does not exhibit pronounced peak and the overall DOS appearance is dominated by the Dirac band (CT1-B1) showing a hexagonal pattern in the real-space.

Figure R2-3 (Supplementary Fig. 11) DFT-calculated LDOS and experimental dI/dV measurements of state $S0$ (-0.2 V) in CT-MoTe₂ phase. (a,b) Magnified dI/dV spectrum of the in-gap states (a) and total DOS (b) of the CT-MoTe₂ phase (c) Theoretical band structures of the CT-MoTe₂ monolayer. The state $S0$ is highlighted by the green color in (a-c). (d) Constant-current dI/dV map at -0.2 V. (e) Theoretically simulated maps derived from the wavefunction norms of the states sitting at -0.2 eV.

Action 2-2-(3):

We added Fig. R2-3 d and e in Fig. 4, put the above discussion into Supplementary Fig. 11 and added sentences of “Spatial maps of states e-S0 to e-S3 ... in Supplementary Fig. 11.” in the second paragraph on page 12.

Comment 2-2-(4): *This is also related to the claim in lines 253-254 on p. 11: “Highly localized charge density... (red circles in Fig. 4f)”, with which I disagree; other features not circled in red show similar magnitudes in Fig. 4f. The authors should provide a comment and explanation for this.*

Reply 2-2-(4):

We are grateful to the Reviewer for this constructive comment. After a closer examination on our data, we agree with the Reviewer. We replotted Fig. 4b and 4f (shown in Fig. R2-4a and R2-4b), while Fig. R2-4b was replotted in Fig. R2-4c using an appropriate color scalebar. Although the state appears stronger around $T_{\text{EMTTD-R}}$, it was not evidenced that it is a highly localized state.

Figure R2-4. Charge density in the constant-current dI/dV map of CT-MoTe₂ acquired at 0.28V. (a) STM topography image of CT-MoTe₂ phase. (b) Constant-current dI/dV maps of (b) acquired at 0.28 V. (c) Images of (b) in an adjusted color scalebar.

Action 2-2-(4):

We revised the sentence of “Highly localized charge density ... (red circles in Fig. 4f)” in the second paragraph on page 12 to “... more pronounced charge density was found dominantly around $T_{\text{EMTTD-R}}$ (the red circle) than that around $T_{\text{EMTTD-B}}$ (the blue circle)” in the second paragraph on page 11. We also replaced Fig. 4g (original Fig.4f) with Fig. R2-4c.

Comment 2-3: *Lines 273-274, p. 11: The authors should elaborate on the claim “Boundary DB-IV, separating two adjacent domains. ... barrier resisting carrier flowing”. I don’t really understand what is meant by this.*

Reply 2-3:

We thank the Referee for raising this valuable question. A schematic diagram is supplied to illustrate that the domain boundary (DB-IV) could be assumed as the transport barrier for charge carriers in the atomic-lattice, while it is “transparent” for the charge carriers in the Te pseudo-sublattice, as depicted in Figure R2-5 (Supplementary Fig. 13).

Figure R2-5 (Supplementary Fig. 13) Schematic diagram of the domain boundary (DB-IV) assumed as the transport barrier or “transparent” for the charge carriers. (a,b) STM topography image of the translational symmetry broken in atomic-lattice (a) and the preserved translation symmetry in Te pseudo-sublattice (b). (c,d) Schematic atomic structure models of DB-IV in the atomic-lattice (c) and Te pseudo-sublattice (d). (e,f) Schematic of the DB-IV behave like a transport barrier (e, in the atomic-lattice) or “transparent” for the charge carriers (f, in the Te pseudo-sublattice).

Action 2-3:

We revised the sentence “Boundary DB-IV ... resisting carrier flowing.” in the first paragraph on page 12 to “The energy-dependent continuity ... as more clearly illustrated in Supplementary Fig. 13.” and added Supplementary Fig. 13 for better illustration.

Minor comment:

Comment 2-4: I understand that in the DFT calculations U was fixed such as to reproduce the MoTe_2 bandgap. It would be good if the authors could justify this value of U , e.g., is it the results of the intrinsic Mo d states? Or of the lattice geometry of the MoTe_2 polymorph, which, via the quasi-flat bands, could give rise to significant electronic correlations? Are similar values of U used in prior DFT studies of MoTe_2 ?

Reply 2-4:

We thank the Reviewer for asking. Here, we believe that the MoTe_2 domains confined

within MTB triangular loops and their resulting electronic (quasi-) flat bands play a paramount role in leading to the on-site Coulomb repulsion, while intrinsic Mo d states play a minor role. The used U value of 1.5 eV is comparable with those used in the literature for T_d -MoTe₂ (2.05eV, *Sci. Adv.* 2021; 7: eabd9275) and Mo₅Te₈ (2.0 eV, *2D Mater.* 2021; 8 015006). In addition, we also added Supplementary Fig. 4 (Fig. R2-6) to show the U -dependence of electronic band structures.

Figure R2-6 (Supplementary Fig. 4) Electronic band-structure and DOSs with different Hubbard U . The U values are 0 eV(a), 1 eV (b), 1.5 eV (c) and 2 eV (d), respectively. In the upper panels, with the increase of U , CT1 bands move up while CT2 bands move down, bringing about a gradually increasing energy gap between t-S0 and t-S1 shown in the lower panels.

Action 2-4:

We added a sentence of “Details of U -dependence on the electronic band structures were discussed in Supplementary Fig. 4.” in the second paragraph on page 7 and added Supplementary Fig. 4 to show the U -dependent band-structures and associated DOSs.

Reviewer #3 (Remarks to the Author):

Manuscript “Electronic Janus lattice and Kagome-like bands in coloring-triangular MoTe₂ monolayers” by Lei and coworker reports a theoretical design and experimental realization of a coloring-triangular (CT) lattice in Te deficient MoTe₂ monolayer; in particular, a Mo₅Te₈ monolayer. This work is the first indication that the Mo₅Te₈ monolayer is a CT lattice, which exhibits a novel electronic Janus lattice and several sets of kagome-like bands. The theoretical and experimental results are of very high quality and well consistent, and the found CT-MoTe₂ exhibiting interesting Janus lattices and kagome bands is indeed intriguing. To the best of my knowledge, this work is also the first realization of the CT lattice in 2D monolayers and, perhaps, in all crystals. It also demonstrated the ability of tuning intralayer domain size and boundary, rather than interlayer twisting and moiré potential in hetero-bilayers, to engineer properties of TMDs. Overall, this is an exciting piece of work in the field of TMD materials and I could see it large impact in the 2D and kagome communities. Before I could recommend it for publication in Nature Communications, there are several points that the authors should clarify.

We thank the Reviewer for his/her careful review, constructive comments, and the positive recommendation.

Comment 3-1: *The authors mentioned “this series of images shows two apparent lattices” in Fig. 3e-3h. Is it possible to distinguish these two lattices in the FFT image of STM topography images or dI/dV conductance maps?*

Reply 3-1:

We thank the Referee for asking this valuable question. As suggested by the Reviewer, we supplied the FFT images of four STM topographic images in Fig. R3-1 (Supplementary Fig. 9). It is noted that the FFT points of the small Te pseudo-sublattice are more pronounced (highlighted by blue circles) in Fig. R3-1e and Fig. R3-1h. In Fig. R3-1g, the FFT pattern of large atomic-lattice is clearly resolved.

Figure R3-1 (Supplementary Fig. 9) FFT of bias-dependent STM images of the CT-MoTe₂ phase. (a-d) Bias-dependent STM topography images of the CT-MoTe₂ phase, showing an apparent electronic Janus lattice. (e-h) The FFT images of corresponding STM image, which can show the atomic-lattice and Te pseudo-sublattice, respectively. It is noted that the FFT points of the small Te pseudo-sublattice are more pronounced (highlighted by blue circles) in (e) and (h).

Action 3-1:

We added a sentence “These two lattices were more straightforwardly illustrated in the FFT images (Supplementary Fig. 9) for those topographic ones shown in Fig. 3e-3h.” on page 10 and added Supplementary Fig. 9 to show the FFT in the revision.

Comment 3-2: *The pseudo-lattice is clearly observable under high bias voltages, i.e. below -1 V or above 1 V. Does it have any intrinsic reason? If the pseudo-lattice dominates in a wide range of energy, it would be curious to know whether the electronic band structures follow the Brillouin zone of the atomic lattice or the pseudo-lattice?*

Reply 3-2:

We thank the Reviewer for raising this insightful issue. We found that the Mo *d* states dominate at higher bias voltages, i.e. below -1 V or above 1 V. These states do not significantly contain Te *p_z* states that distributed following the atomic-lattice in the real-space. Preliminary ARPES results (not shown here) also support this feature that some bands indeed follow the BZ of the Te pseudo-lattice.

Action 3-2:

Since these results are preliminary and not directly relevant with the present work, we will be working on this topic in future.

Comment 3-3: *It would be interesting to briefly explain why the domain boundary displayed in Fig. 5a is invisible at certain bias voltages. Is it relevant to a unique electronic state? Does it have application potentials in further electronic devices?*

Reply 3-3:

We thank the Referee for this valuable question. The CT-MoTe₂ exhibits electronic Janus lattices, featuring the atomic-lattice at low bias and the Te pseudo-sublattice at high bias. When CT-MoTe₂ shows the Te pseudo-sublattice, translation symmetry is preserved at DB-IV, as depicted by the green dashed line in Fig. R3-2a and corresponding atomic structure in Fig. R3-2b. Conversely, when CT-MoTe₂ shows the atomic-lattice, translation symmetry is broken at DB-IV, as illustrated by the red dashed line in Fig. R3-2c and corresponding atomic structure in Fig. R3-2d. Thus, the domain boundary was nearly indistinguishable due to the preserved translation symmetry at the Te pseudo-sublattice, but became distinctly observable as a result of the broken translation symmetry at atomic-lattice. It suggests potential applications in future electronic devices, as we have briefly shown in Fig. R2-5 e and f.

Figure R3-2. Schematic of the domain boundary in different bias voltages. (a,b) STM topography image (a) and corresponding atomic structure (b) of the domain boundary with the almost indistinguishable domain boundary. (c,d) STM topography image (c) and corresponding atomic structure (d) of the translation symmetry broken at atomic-lattice.

Comment 3-4: *Kagome lattice is a well-known lattice, but CT is recently proposed one which is not as familiar as the kagome lattice to the 2D community. Could the authors*

comment on the connection(s) between the kagome and CT lattices?

Reply 3-4:

We thank the Reviewer for this suggestion. By rotating the triangles following the way illustrated in Fig. R3-3a and R3-3b in the kagome lattice, one could obtain the CT lattice (Fig. R3-3c). The CT-lattice, sharing with the kagome lattice, hosts a hexagonal lattice and a triangle on each of its nodes, which indicates the CT lattice is inherently equivalent to the kagome lattice.

Figure R3-3 (Supplementary Fig. 3) Line graph of kagome lattice and CT lattice. Line graph of a kagome lattice (a), distorted kagome lattice (b) and CT lattice (c).

Action 3-4:

We added a sentence “The relationship between the kagome and CT lattices was illustrated in Supplementary Fig. 3.” in the second paragraph on page 5 and added Supplementary Fig. 3.

Comment 3-5: As mentioned in the methods section, the CT-MoTe₂ monolayer was prepared by controllably desorption of Te atoms from the MoTe₂ monolayer. Could the authors clarify the route(s) to monitor and control the desorption of Te accurately?

Reply 3-5:

We thank the Referee for raising this professional question. During the deposition process, beam flux monitor (BFM) can be utilized to swiftly determine flux ratios and growth rates in proximity or at the sample position. This enables a rapid assessment of flux ratios from different sources prior to growth, such as Mo atoms and Te atoms in the CT-MoTe₂ case. During annealing, reflection high-energy electron diffraction (RHEED) observations reveal that stripe distances deviate from those of MoTe₂ phase. As a new stripe gradually forms, synthesis of a new phase occurs. The sample was monitored by BFM and RHEED to regulate the temperature of both the source and annealing process to form CT-

MoTe₂ phase.

Action 3-5:

We added a sentence of “The sample was monitored by BFM and RHEED to regulate the temperature of both the source and annealing process to form CT-MoTe₂ phase.” in the methods.

Comment 3-6: *If the Te desorption process is precisely controllable, could one get a monolayer comprised of those N=3 or N=6 uniform MTB triangles? If yes, which structures are energetically more favored? Are they experimentally accessible?*

Reply 3-6:

We thank the reviewer for this inspirational comment. We theoretically constructed many N=1, 3, and 6 MTB superlattices and compared their formation energies (Fig. R3-4a). We found a N=3 MTB superlattice (Fig. R3-4b) of superior stability, which was occasionally observable in our experiments (Fig. R3-4c), and was very recently observed by another group (arXiv:2307.06001). In addition, the calculations did not consider substrate effects which might change the relative stability of these phases, leading other phases to be energetically favored. These results are not directly relevant with the present work. They are thus for review only and are going to be discussed in details elsewhere.

Figure R3-4. Formation energies of MTB superlattices. (a) DFT calculated formation energies of N=1, 3, and 6 MTB superlattices as a function of Te chemical potential μ_{Te} . T0-T3 represent the number of blue Te atoms in (b). They simulate the uncertainty in experimental growth parameters (i.e., the Te-deficient and Te-rich conditions). (b) Structural model of N₆-T1 with a “CT-like” lattice. (c) STM of N=3 MTB superlattice, where the Te_{MTTD-R} region is marked by red dashed triangle. (c) $V=1.1$ V, $I=100$ pA.

Comment 3-7: *The post-growth annealing method aside, are there any other likely routes to synthesize this CT-MoTe₂ monolayer? If yes, could these methods applicable to other TMD materials like MoSe₂, NbTe₂, TaTe₂?*

Reply 3-7:

We thank the Referee for this constructive question. This is an open question that we cannot exactly and precisely answer so far, but we could infer the possibility and feasibility of other methods for constructing the CT lattice as follows.

(1) Besides the post-growth high-temperature annealing (method 1: Te desorbing) method in this work, the CT lattices may also be induced by post-synthesis modification by incorporation of extra molybdenum atoms (method 2: Mo doping) to introduce high-density MTB to form CT-MoTe₂. This method has been used in [ACS Nano 12, 3975-3984 (2018)] to generate high-density MTB, indicating its potential for constructing the CT lattices.

(2) Both methods could be applicable to the TMD materials to realize CT-MoSe₂, CT-MoS₂ and WSe₂. According to our knowledge, the NbTe₂/TaTe₂ mostly exist in the phase of 1T, while the MTB formed in the H-phase of TMD. In our opinion, there is very small possibility to synthesis the CT-NbTe₂/CT-TaTe₂ phase, while more experimental and theoretical works are needed to discuss this interesting question.

There are some minor technical issues, see comments below:

Comment 3-8a: *There are three labels, namely 1T', 1H and MTB, marked in Fig.1b. I wonder if the domain labeled using MTB appropriate or should it be presented by another label?*

Reply & Action 3-8a: We are sorry for the confusing labels. The domain we labeled as “MTB” contained both MTB and H-MoTe₂. We carefully modified them from “MTB” to “MTB & 1H” to ensure they are clear enough to the readers.

Comment 3-8b: *Term “Te_{MTTD-R}” was used to identify a certain group of Te atoms. Could the authors provide a brief explanation to this term?*

Reply & Action 3-8b: We thank the Referee for this reminder. We added corresponding explanation in the manuscript: “Te_{MTTD-R} atoms are highlighted by red balls”

Comment 3-8c: *In the figure caption of Fig.4, it says “... The Te_{MTTD-R} and Te_{MTTD-B} atoms are marked by the red and blue circles in (b, d-l), respectively...”. Here, panel “d-l” should, perhaps, read as panel “e-j”.*

Reply & Action 3-8c: We overlooked and the Reviewer is correct. We added two panels in the revised Fig. 4 and changed “b, d-l” to “a, e-l” in the revised caption.

REVIEWERS' COMMENTS

Reviewer #1 (Remarks to the Author):

I would like to thank the authors who made dedicated efforts in addressing all reviewers' comments. I am mostly satisfied with authors' responses and revisions to my previous review. However, I still have a couple of minor comments for authors to consider in finalizing their manuscript.

(1) In response to my comment on p/d-orbital symmetry in affecting flatness of flat band, I appreciate the additional detailed calculations and symmetry analysis. I would like to point out that there is an intuitive way to show that perfect flat band can be obtained by rotating p/d-orbital to restore C_3 rotation symmetry in a Kagome lattice [see the very recent work by Kim and Liu, PRB 107, 205130 (2023)]. The authors' detailed calculation and analysis is likely consistent with this simple physical picture.

(2) In response to the Reviewer #3's comment on the relationship between Kagome and CT. The authors may point out that in the original proposal of CT lattice [Phys. Rev. B., 99, 100404(R) (2019)], it was shown that the CT lattice Hamiltonian is related to the Kagome lattice Hamiltonian by a unitary matrix of rotation operator (representing rotation of two triangles as also shown by figures plotted by the present authors). Therefore, the two lattice Hamiltonians must have identical eigenvalues or band structures.

Reviewer #3 (Remarks to the Author):

In this revised manuscript, the authors have properly addressed all the concerns and suggestions raised by the present referee and referee #2. Hence, I would like to recommend its publication with highest priority for timing consideration.

REVIEWER COMMENTS

Reviewer #1 (Remarks to the Author):

I would like to thank the authors who made dedicated efforts in addressing all reviewers' comments. I am mostly satisfied with authors' responses and revisions to my previous review. However, I still have a couple of minor comments for authors to consider in finalizing their manuscript.

We thank the Reviewer for his/her careful review, constructive comments, and the positive recommendation.

Comment 1-1: *In response to my comment on p/d -orbital symmetry in affecting flatness of flat band, I appreciate the additional detailed calculations and symmetry analysis. I would like to point out that there is an intuitive way to show that perfect flat band can be obtained by rotating p/d -orbital to restore C_3 rotation symmetry in a Kagome lattice [see the very recent work by Kim and Liu, *PRB* 107, 205130 (2023)]. The authors' detailed calculation and analysis is likely consistent with this simple physical picture.*

Reply 1-1:

We thank the Reviewer for bringing about this relevant publication, which were properly cited in the revision.

Action 1-1:

We added “which was demonstrated by a tight-binding (TB) d -orbital kagome lattice model intuitively⁴⁴” in the first paragraph of page 7 in the manuscript.

Comment 1-2: *In response to the Reviewer #3's comment on the relationship between Kagome and CT. The authors may point out that in the original proposal of CT lattice [*Phys. Rev. B.*, 99, 100404(R) (2019)], it was shown that the CT lattice Hamiltonian is related to the Kagome lattice Hamiltonian by a unitary matrix of rotation operator (representing rotation of two triangles as also shown by figures plotted by the present authors). Therefore, the two lattice Hamiltonians must have identical eigenvalues or band structures.*

Reply 1-2:

We thank the Reviewer for this enlightening comment. We concur with the Reviewer's comment regarding the necessity of elucidating the connections between Hamiltonian in CT lattice and Kagome lattice to enhance our perspective.

Action 1-2:

We added the sentence “It has also been proven that the CT lattice is equivalent to the renowned kagome lattice mathematically by a unitary transformation [1]. The CT lattice Hamiltonian is shown to be connected to the Kagome lattice Hamiltonian through a unitary matrix of rotation operator, indicating that the two lattice Hamiltonians possess identical eigenvalues or band structures.” in the caption of Supplementary Fig. 3.

Reviewer #3 (Remarks to the Author):

In this revised manuscript, the authors have properly addressed all the concerns and suggestions raised by the present referee and referee #2. Hence, I would like to recommend its publication with highest priority for timing consideration.

We express our sincere gratitude to the Reviewer for his/her meticulous review of our revised manuscript and wholehearted recommendation.